# The stigma of alcohol use during pregnancy: Exploring the impact of alcohol strength and social context on public perceptions

Sam Burton[1,2]*, Shannon Cummings[2], Charlotte Connides-Smith[1], Elizabeth Fairweather[2], Catharine Montgomery[1], Abigail K. Rose[1], Poppy Whalley[1], Andrew Jones[1]

1 School of Psychology, Liverpool John Moores University, Liverpool, United Kingdom, 2 Department of Women's and Children's Health, School of Life Course and Population Sciences, Faculty of Life Sciences & Medicine, King's College London, London, United Kingdom

* S.P.Burton@ljmu.ac.uk

## Abstract

Alcohol use during pregnancy remains a major public health concern, with societal expectations of abstinence often resulting in stigma toward women who drink. To examine stigma and discrimination attributed to pregnant women consuming alcohol of varying strength and social context, within hypothetical scenarios. We explored whether greater stigma was attributed to consuming alcohol in social settings vs. alone, and whether consumption of low and alcohol-free drinks are less stigmatised than standard strength alcohol drinks. The research employed a vignette-based experimental design across three studies, involving hypothetical social scenarios depicting alcohol consumption during pregnancy. Participants were randomly assigned to alcohol (3 levels: no, low and standard alcohol) and social context (2 levels: alone or with friends) conditions recruiting a total of 1,054 participants. Measures of stigma were assessed using the Stigma and Attribution Assessment (SAA) and Personal and Perceived Public Stigma Measure (PPSM). Discriminatory behaviour was measured in two studies using a novel discrimination task. Pregnant women consuming standard and low-alcohol drinks were more stigmatised than those consuming alcohol-free beverages across multiple aspects of stigma. Pooled analyses showed that standard alcohol was associated with greater stigma on measures including social distance, perceived danger, prognostic optimism, public stigma, and personal discrimination. Low-alcohol drinks were also more stigmatised than alcohol-free drinks across domains such as blame, continued care, public stigma, and treatment stigma. The stigma attributed to women consuming alcohol during pregnancy may vary as a function of alcohol strength, with consumption of alcohol-free products associated with least stigma. Stigma attribution seemed driven by societal norms around abstinence. Public health messaging should shift towards risk-informed education to shape societal norms of abstinence and reduce stigmatisation.

**Data availability statement:** Availability of data and materials: Data is available from https://osf.io/4c825.

**Funding:** The author(s) received no specific funding for this work.

**Competing interests:** The authors have declared that no competing interests exist.

## Background

Alcohol consumption during pregnancy is a leading cause of preventable birth defects and intellectual disabilities [1]. The UK guidelines around drinking in pregnancy (updated in 2016) recommending abstinence. However, it is estimated that the global and UK prevalence rates of alcohol consumption during pregnancy are 9.8% and 41.3%, respectively [2]. While detrimental effects of heavy alcohol use are well documented, evidence around low or moderate consumption is inconsistent, yet there is evidence of adverse maternal and child outcomes [3,4].

In order to develop more effective, tailored alcohol interventions to reduce alcohol exposed pregnancy, it is also important to understand why women remain abstinent during pregnancy and why they do not seek help for alcohol use during pregnancy [5]. Stigma involves the co-occurrence of labelling, stereotyping and cognitive separation into "us" and "them" groups, resulting in power imbalances, social rejection and discrimination [6]. Evidence suggests that women choose not to drink due to the potentially harmful effects to the baby and the guilt of breaking societal norms around abstinence in pregnancy which may result in experiencing stigma [7,8]. Stigma involves a series of socially constrained actions that recognise, separate, and disempower individuals with attributes deemed unacceptable by society, manifesting at three levels: self, social, and structural stigma [9,10]. Self-stigma occurs when individuals internalize and apply societal beliefs to themselves, while social stigma reflects the psychological, emotional, and behavioural responses of non-stigmatised individuals [10,11]. Structural stigma operates through societal customs and organizations, further intertwining with social stigma in settings such as healthcare, especially in cases of perinatal substance use, where intersecting stigmas regarding addiction and substance use create additional challenges [12,13]. Research shows that stigma can be a barrier to women disclosing alcohol use during pregnancy [14].

To add to the complexity, norms around not drinking in pregnancy can come into conflict with social expectations to consume alcohol and within certain contexts [8]. Evidence identifies special occasions and social contexts to be drivers of alcohol use across a range of countries and cultures, with self-reported alcohol use during pregnancy increasing by 33.3% when asked about drinking in social contexts [15,16]. Women report feeling they must drink before and after pregnancy disclosure often as a result of social interactions in which alcohol consumption is encouraged, e.g., by family and peers within group settings [17,18]. In general populations, social support has been shown to influence risk drinking, both positively and negatively [19,20]. Pregnant women often report drinking with friends and family [21], suggesting that social context may influence decisions around alcohol consumption. However, the role of social context in shaping these decisions remains underexplored, particularly how social context and norms may impact alcohol use during pregnancy and associated stigma [18]. Research highlights a need for public health messaging to challenge social pressures to drink, shifting cultural expectations towards abstinence in pregnancy [22,23].

Women who consume alcohol during pregnancy often encounter social stigma in the form of negative stereotypes [13], they can be seen as deviating from the

'motherhood ideal' and labelled as 'bad mothers' [24]. A recent review Lyall, Wolfson [18] highlighted how women who drink during pregnancy regularly experience discriminatory attitudes at an interpersonal level, with degree of stigmatisation varying based on alcohol consumption patterns [7]. Structural stigma extends into healthcare services through the stigmatising attitudes of healthcare staff [13,25], discouraging women from seeking treatment and leading them to adopt strategies to evade detection [26] in turn leading to a reduction in access to appropriate treatment.

While drinking typical alcohol drinks (above 1.2% ABV, e.g., not a low or no alcohol drink) may be stigmatised and unacceptable during pregnancy and deemed as a prerequisite of being a "good mother" [7,27,28], recent advancements in the alcohol-free beverage market [29] require further investigation in respect to stigma and acceptability during pregnancy (Burton et al., 2025). NoLo products may be deemed as more acceptable, we must understand the wider impacts given there is no known safe level of alcohol consumption [30]. Currently there are several categories of 'NoLo' drink products, e.g., alcohol-free (≤0.05% ABV), de-alcoholised (0.05–0.5% ABV), and low alcohol (≤1.2% ABV). The alcohol industry is also diversifying with an increase in alcohol products with a lower alcohol beverage volume (ABV) than the 'standard'(e.g., 1.2% ABV or less) [31]. Lower than standard ABV drinks may be consumed as a proxy for NoLo drinks rather than abstinence [32], this is primarily due to poor knowledge around categorisation definitions [33]. Within the context of pregnancy, NoLo and lower than average alcohol products may be stigmatised less than standard products, which in turn may enable alcohol use against abstinence recommendations [34].

Women who consume alcohol during pregnancy are stigmatised by society through the motherhood ideal, and the societal view that the priority of women is to protect their baby [35]. While research has shown that low and moderate alcohol consumption is viewed as acceptable in some circumstances [17], no work to our knowledge has examined No-Lo alcohol drinks and the attributed stigmatisation. Given the increasing marketisation of NoLo products [29] it is crucial to understand the stigma attributed towards their use, given how recent work suggests women may still experience guilt and shame following consumption of NoLo products, and that such products aid in management of stigmatisation but in turn may also create stigma by appearing as standard alcohol products [36]. As such, the current research consisted of three studies which used a vignette (scenario) method to determine stigma attributed to alcohol use during pregnancy (with different measures of social stigma). Based on prior work on the 'motherhood ideal', we hypothesised (H1.) that level of stigma attributed to women consuming 'alcohol' during pregnancy would vary as a function of alcohol strength (ABV%), with greatest stigma associated with standard strength wine and least stigma association with alcohol free wine. We also hypothesised (H2.) that level of stigma would be greatest towards pregnant women consuming alcohol alone rather than in a social environment.

## Methods

### Design and procedure

All three studies adopted a 2 (social context: alone vs. with friends) x 3 (alcohol type: standard ABV vs low ABV vs alcohol free) between-participant design. Participants were recruited via research participation schemes (SONA Systems Ltd.), Prolific Academic (https://prolific.co/; see [37] and social media platforms (e.g., Twitter, Facebook). All participants were aged over 18 years of age, were fluent in English, not currently pregnant, no current or former diagnosis of a substance use or eating disorder.

In each study participants were instructed to complete the experiment in a quiet space without distractions (self administered). After providing informed consent, participants were randomly assigned (via Inquisit [v.t Millisecond, Seattle] which hosted the studies) to one of the six vignette conditions, which remained on the screen for a minimum of 60 seconds. Participants completed the Stigma and Attribution Assessment (SAA), Personal and Perceived Public Stigma Measure (PPSM) and the financial discrimination task, finally they completed the demographics questions, and the timeline follow back. Participants were asked to answer the SAA and PPSM in respect to individual (Sarah) presented within the vignette, to measure the stigma attributed towards the individual (Sarah) as a result of the vignette read by the participant.

Ethical approval was granted by the School of Psychology Research Ethics Committee at the host university (reference number REDACTED).

## Participants

Across the three studies, 1054 participants were recruited from the UK to take part in an online study, see Table 1 for demographics. In study 1 we recruited 179 participants; sample size was not pre-defined a priori but we used the effects from study 1 to power subsequent studies. Study 2 was powered based off simulation analysis for a 2x3 between subjects design, using Superpower package in R. Based off 90% power, alpha=0.05 it was estimated we would need 64 participants per group (target N=384, actual N=388). In study 3 we powered based on simulation analysis of a main effect of alcohol on the financial discrimination task in study 2. We estimated with 90% power, alpha=0.05 we would need 81 participants per group (target N=486, actual N=487). Recruitment began on the 24/2/2024 and ended on the 4/11/2024.

## Measures

**Demographics.** Age, gender identity, highest level of education and average household income (before tax).

**Table 1. Participant characteristics presenting n(%), mean (standard deviation), median (inter-quartile range), across all three studies and in total.**

|  | Study 1 (n=179) | Study 2 (n=388) | Study 3 (n=487) | Total (N=1054) |
|---|---|---|---|---|
| **Age** |  |  |  |  |
| 18-25 | 113 (63.13%) | 65 (16.75%) | 82 (16.84%) | 260 (24.67%) |
| 26-35 | 15 (8.38%) | 134 (34.54%) | 148 (30.39%) | 297 (28.18%) |
| 56-65 | 19 (10.62%) | 32 (8.25%) | 42 (8.62%) | 93 (8.82%) |
| 36-45 | 20 (11.17%) | 83 (21.39%) | 113 (23.20%) | 216 (20.49%) |
| 46-55 | 9 (5.03%) | 64 (16.50%) | 77 (15.81%) | 150 (14.23%) |
| 65+ | 3 (1.68%) | 10 (2.58%) | 25 (5.13%) | 38 (3.61%) |
| Missing | 0 | 0 | 0 | 0 |
| **Income** |  |  |  |  |
| Mean £ (SD) | 44,178.77 (42,339.21) | 47,231.32 (29662.49) | 44,017.32 (30,137.61) | 45,230.07 (32,422.63) |
| Median £ (IQR) | 31,000 (47,500) | 40,000 (34,000) | 39,000 (31,000) | 40,000.00 (36,000) |
| **Gender n(%)** |  |  |  |  |
| Female | 150 (83.80%) | 376 (96.91%) | 477 (97.95%) | 1003 (95.16%) |
| Male | 22 (12.29%) | 5 (1.29%) | 6 (1.23%) | 33 (3.13%) |
| Non-binary | 5 (2.80%) | 7 (1.81%) | 4 (.82%) | 16 (1.51%) |
| Missing | 2 (1.12%) | 0 | 0 | 2 (.20%) |
| **Education n(%)** |  |  |  |  |
| Primary school | 1 (.56%) | 0 | 0 | 1 (.10%) |
| Secondary school | 24 (13.41%) | 122 (31.44%) | 160 (32.85%) | 306 (29.03%) |
| Degree | 11 (6.15%) | 185 (47.68%) | 208 (42.71%) | 404 (38.33%) |
| Postgraduate | 0 | 79 (20.36%) | 115 (23.61%) | 194 (18.41%) |
| Other | 5 (2.79%) | 2 (.52%) | 4 (.83%) | 11 (1.04%) |
| Missing | 138 (77.14%) | 0 | 0 | 138 (13.09%) |
| **Weekly units consumed** |  |  |  |  |
| Mean (SD) | 22.83 (31.21) | 12.58 (22.75) | 9.95 (19.53) | 13.10 (23.49) |
| Median (IQR) | 13.00 (31.00) | 6.00 (16.00) | 3.00 (12.00) | 4.00 (16.00) |

Table 1 n(%) and mean (standard deviation) for age, income, gender, education and weekly units of alcohol consumed.

**Vignette/Scenario.** In all studies, a short scenario described a woman called Sarah attending her friend's wedding reception. The atmosphere was described as exciting and fun, with Sarah enjoying spending time at the event. The scenario explains that Sarah decides to go to the bar to get drinks, participants are randomised to one of two conditions where Sarah is either alone or with friends, and chooses to get a glass of sparkling wine to celebrate. The vignette then introduces the information that Sarah is pregnant. The vignette states that Sarah asks the bar person if there is sparkling wine available. At this point, participants are randomised to one of three drink availability conditions: standard, low alcohol, and alcohol free. In each condition, Sarah decides to accept the drink that is available. (See supplementary file 1 for example vignette)

**Stigma and Attribution Assessment (SAA).** The SAA [38] assessed multiple dimensions of stigma towards problematic substance use. The 22-item questionnaire comprises five subscales including prognostic optimism (e.g., Sarah will be able to maintain abstinence over the next three months), perceived danger (e.g., I believe Sarah is dangerous), social distance (e.g. I'd be happy to have Sarah as my neighbour), blame attribution (e.g. Sarah's alcohol use is definitely genetic in origin) and need for continuing care (e.g., Sarah will need prolonged support for their alcohol use). Kelly et al. (2021) and Pennington, Monk [39] found all subscales to have internal reliability ($\alpha > .68$), the present study found them to be acceptable ($\Omega > .70$). Higher scores corresponded to greater danger and continued care, whereas lower scores correspond to lower blame and prognostic optimism along with greater social distance. Higher scores correspond to greater danger and continued care, while lower scores correspond to greater social distance, lower blame and prognostic optimism.

**Personal and Perceived Public Stigma Measure (PPSM).** The PPSM [40] assessed public stigma. The PPSM consists of 23-items with four subscales including perceived public stigma (e.g., People like them should feel embarrassed about their situation), perceived treatment stigma (e.g., Opportunities would be limited if people knew they received treatment), personal stereotype/prejudicial stigma (e.g., How likely is it they would do something violent to themselves), and personal discriminatory stigma (e.g., I would be willing to befriend them). Rundle et al. [40] and Pennington et al. [39] found all subscales to have internal reliability ($\alpha > .69$), the present study found them to be acceptable ($\Omega > .73$). Responses were on a scale of 1–4 (lower to higher endorsement) and summed to create a total score. Higher scores correspond to greater stigmatising perceptions (Holman [41]).

**Discrimination task.** A novel discrimination task [42,43] assessed discriminatory behaviour (financial in study 2 or social in study 3). The task presents a fictional cognitive training website called "Psy-Learn" in which the participant acts as a supervisor to individuals who are enrolled. In this task participants are shown the individual enrolled 'performance' on a series of cognitive trials, such as a word anagram, memory test, and simple reaction time task. After they observe this performance on each trial, they are told the learner was correct or incorrect (there were three instances in the learner was correct and incorrect). Participants are then required to provide a reward or punishment to the learner depending on whether they were correct or incorrect, respectively. In this case, the learner was the individual described in the hypothetical vignettes ('Sarah') below.

In study 2, participants were asked to reward or punish Sarah financially (reward performance on each trial by giving 0 to 100 pence, or punish performance on each trial by removing 0–100 pence).

In study 3 the reward/ punishment was a social reward (a sliding scale representing 0–100 from a neutral to a happy face on reward trials, or a neutral to an unhappy face on punishment trials). Each participant, irrespective of condition, saw identical learner performance across all trials.

Two dependent variables are computed from the task: reward summed across the three correct answers (0–300) and punishment summed across the three incorrect answers (0–300). Lower rewards and greater punishment correspond to great discriminatory behaviour, respectively. Previous research has suggested that individuals from stigmatised groups (e.g., individuals living with obesity, individuals who use substances) are more likely to be discriminated against [42,43]. Following the presentation of the six trials participants were then also asked 'Overall, would you recommend Sarah be

permitted to continue to the next stage of the PSY-LEARN program" with a binary option (YES, NO), which provides an outcome similar to denial of service, used in hypothetical stigma paradigms, e.g., Swami and Monk [44]. Previous research has demonstrated that this task is sensitive to discrimination behaviours towards stigmatised groups, e.g., individuals with obesity or a substance use disorder [42,43]

**Current alcohol consumption.** The Timeline Followback (TLFB, [45]) assessed weekly alcohol use. Using a daily diary format, participants were asked to record how many and what type of drink (e.g., large/small glass of wine, pint of beer) they had consumed over the past 14 days. Drinks were converted to units (1 UK unit = 8 g alcohol) and an average was calculated for weekly alcohol unit consumption.

## Analysis

Data were analysed using R studio with the 'tidyverse' [46], ''ggplot' (Wickham, 2016), 'effectsize' [47], and 'psych' [48] packages. Any income greater than what was deemed implausible by examination of the distributions and box plot was also removed (10 participants reporting income of above £270,000). These participants were retained for the inferential analysis. Between subject ANOVAs were used to test for main effects and interactions in terms of social context (2 levels: alone or with friends) and alcohol type (3 levels: no, low and standard alcohol), in all studies for the dependent variables of SAA, PPSM and financial discrimination task (for study 2 and 3). Post-hoc pairwise comparisons were conducted using Tukey's Honestly Significant Difference Test to explore significant effects. *P*-values were adjusted for multiple comparisons using Tukey's adjustment.

## Results

### Descriptives

See table 1 for descriptive statistics of the demographic information collected across studies.

### Study 1

Table 2 shows mean and standard deviation of stigma and discrimination variables, split by levels of the independent variable (context and alcohol strength) across the studies.

**PPSM.** There was no significant main effect of alcohol type ($F_{(2, 136)} = .16$, $p = .850$, $np^2 = .00$) or social context ($F_{(1, 136)} = .04$, $p = .843$, $np^2 = .00$) on PPSM total score or either of the four subscales ($p's > 266$).

**SAA.** There was no significant main effect of alcohol type ($F_{(2, 172)} = .46$, $p = .632$, $np^2 = .01$) or social context ($F_{(1, 172)} = .51$, $p = .475$, $np^2 = .00$) on the SAA total score or either of the five subscales ($p's > .104$).

### Study 2

**Financial discrimination task.** There was no significant main effect of alcohol type ($F_{(2, 382)} = 2.02$, $p = .134$, $np^2 = .01$) or social context ($F_{(1,382)} = .08$, $p = .772$, $np^2 = .00$) on punishment administered on the financial discrimination task. There was a significant main effect of alcohol type ($F_{(2,382)} = 4.84$, $p < .01$, $np^2 = .02$) on reward administered on the financial discrimination task. Those who consumed standard strength alcohol were rewarded significantly less than the those who consumed alcohol free. There were no other significant contrasts, see Fig 1.

**Denial of service.** Chi squared test demonstrated no significant associations between the alcohol type ($X2(2) = 2.65$, $p = .266$) or social context ($X2(2) = 0.27$, $p = .601$) and denial of service on the discrimination task.

**PPSM.** There was a significant main effect of alcohol type on PPSM total score ($F_{(2, 381)} = 30.22$, $p < .001$, $np^2 = .14$). In addition alcohol type had a significant main effect on public stigma ($F_{(2, 381)} = 33.20$, $p < .001$, $np^2 = .15$), treatment stigma ($F_{(2, 382)} = 11.02$, $p < .001$, $np^2 = .05$), personal discriminatory stigma ($F_{(2, 382)} = 22.54$, $p < .001$, $np^2 = .11$), and personal stereotype treatment stigma ($F_{(2, 382)} = 17.82$, $p < .001$, $np^2 = .09$). Consumption of standard and low alcohol

**Table 2. Mean (Standard Deviation) of PPSM and SAA subscales, and discrimination tasks by alcohol type and social context.**

| Study 1- Measures | Social- No | Social- Low | Social-Standard | Alone-No | Alone-Low | Alone-Standard |
|---|---|---|---|---|---|---|
| Public stigma | 6.56 (1.89) | 7.08 (1.67) | 6.44 (1.83) | 6.33 (1.65) | 6.37 (1.57) | 6.42 (2.17) |
| Treatment | 5.60 (1.63) | 5.35 (1.77) | 5.32 (1.46) | 5.10 (1.45) | 5.67 (1.82) | 5.00 (1.59) |
| Personal Disc | 14.48 (3.34) | 14.69 (2.74) | 15.32 (2.25) | 15.06 (3.14) | 14.78 (2.19) | 14.87 (2.34) |
| Personal Stereo | 220.32 (4.14) | 23.00 (3.72) | 22.79 (3.69) | 23.81 (4.82) | 23.41 (3.32) | 22.73 (3.07) |
| Social Distance | 19.53 (11.46) | 16.40 (9.95) | 15.73 (11.07) | 16.66 (13.01) | 20.94 (10.35) | 17.75 (9.27) |
| Perceived Danger | 11.53 (8.92) | 13.77 (9.36) | 11.50 (8.85) | 9.62 (8.48) | 13.52 (8.28) | 15.86 (9.51) |
| Prognostic Optimism | 16.63 (10.01) | 15.23 (9.13) | 13.27 (9.63) | 14.86 (11.51) | 18.10 (8.72) | 15.43 (8.23) |
| Blame Attribution | 6.53 (4.80) | 5.90 (3.91) | 5.10 (4.37) | 5.48 (4.87) | 6.48 (3.90) | 6.54 (3.86) |
| Continued Care | 1.90 (1.52) | 1.77 (1.50) | 1.90 (1.65) | 1.38 (1.40) | 1.94 (1.50) | 2.29 (1.54) |
| Study 2- Measures | Social- No | Social- Low | Social-Standard | Alone-No | Alone-Low | Alone-Standard |
| Public stigma | 5.19 (2.04) | 6.36 (2.26) | 7.19 (2.71) | 4.98 (2.32) | 6.73 (2.38) | 8.22 (2.96) |
| Treatment | 4.19 (1.61) | 5.25 (1.77) | 5.42 (1.91) | 4.55 (2.13) | 5.17 (1.65) | 5.22 (1.69) |
| Personal Disc | 9.37 (3.76) | 11.71 (4.12) | 12.41 (4.02) | 8.78 (3.65) | 11.27 (4.56) | 12.73 (4.83) |
| Personal Stereo | 16.67 (4.14) | 18.87 (4.47) | 19.54 (3.96) | 16.88 (4.54) | 19.21 (4.87) | 20.98 (5.42) |
| Social Distance | 27.22 (6.02) | 25.24 (6.20) | 23.66 (6.05) | 28.22 (6.31) | 24.30 (6.32) | 21.73 (6.36) |
| Perceived Danger | 11.21 (5.28) | 14.22 (5.998) | 16.16 (5.74) | 10.52 (4.62) | 14.20 (6.21) | 17.80 (6.87) |
| Prognostic Optimism | 24.03 (5.28) | 21.12 (5.98) | 20.69 (5.74) | 24.43 (4.62) | 20.27 (6.21) | 18.65 (6.87) |
| Blame Attribution | 5.63 (2.86) | 6.06 (2.61) | 6.78 (2.38) | 5.64 (2.74) | 6.20 (2.29) | 6.71 (2.52) |
| Continued Care | 1.99 (1.35) | 1.93 (1.10) | 2.85 (1.43) | 1.90 (1.36) | 2.14 (1.13) | 2.84 (1.39) |
| Financial Reward | 264.19 (56.85) | 246.37 (72.65) | 240.95 (71.15) | 267.74 (53.44) | 251.69 (57.07) | 241.08 (72.55) |
| Financial Punishment | 53.82 (57.33) | 62.06 (54.28) | 62.84 (55.20) | 48.00 (52.28) | 57.99 (48.81) | 68.27 (66.18) |
| Study 3- Measures | Social- No | Social- Low | Social-Standard | Alone-No | Alone-Low | Alone-Standard |
| Public stigma | 5.95 (2.51) | 6.75 (2.53) | 6.89 (2.39) | 5.49 (2.41) | 6.36 (2.77) | 8.13 (2.37) |
| Treatment | 5.00 (2.02) | 5.28 (1.88) | 5.65 (1.99) | 4.50 (1.70) | 4.99 (1.71) | 5.57 (1.77) |
| Personal Disc | 10.94 (4.19) | 11.61 (4.34) | 12.71 (4.64) | 10.49 (4.10) | 11.69 (4.44) | 13.35 (4.59) |
| Personal Stereo | 18.17 4.60) | 19.17 (4.17) | 20.10 (4.57) | 17.72 (4.16) | 19.14 (4.32) | 21.62 (4.26) |
| Social Distance | 25.06 (5.86) | 24.25 (5.72) | 23.13 (6.07) | 26.38 (5.94) | 23.99 (5.63) | 21.07 (5.76) |
| Perceived Danger | 13.25 (6.17) | 15.03 (5.86) | 15.80 (5.99) | 12.15 (5.70) | 14.23 (5.40) | 17.66 (7.45) |
| Prognostic Optimism | 22.42 (5.09) | 20.34 (4.83) | 20.58 (4.21) | 23.50 (4.88) | 20.95 (4.81) | 18.66 (4.89) |
| Blame Attribution | 6.42 (2.75) | 6.57 (3.10) | 6.64 (2.71) | 6.06 (3.00) | 6.81 (3.08) | 6.50 (2.35) |
| Continued Care | 2.48 (1.60 | 2.59 (1.57) | 2.64 (1.41) | 2.03 (1.33) | 2.47 (1.54) | 2.81 (1.46) |
| Social Reward | 267.26 (37.37) | 270.44 (36.98) | 265.21 (35.20) | 277.17 (30.456) | 269.04 (43.21) | 272.75 (31.65) |
| Social Punishment | 148.05 (68.31) | 131.52 (57.93) | 140.04 (59.57) | 152.04 (59.42) | 146.19 (59.63) | 148.25 (68.29) |

led to more stigmatising responses than those consuming alcohol-free drinks ($p$'s < .001, see Table 3). Consumption of standard alcohol drinks had higher public stigma, than low and alcohol free ($p < .01$.) There was no significant effect of social context (F(1, 381) = .63, $p = .429$, $np^2 = .00$).

**SAA.** There was no significant effect of social context (F(1, 382) = 2.18, $p = .141$, $np^2 = .01$) on SAA total score, yet there was significant main effect of alcohol type (F(2, 382) = 4.15, $p < .05$, $np^2 = .02$). Alcohol type had a significant main effect on the SAA subscales of social distance (F(2, 382) = 19.16, $p < .001$, $np^2 = .09$), perceived danger (F(2, 382) = 32.90, $p < .001$, $np^2 = .15$), prognostic optimism (F(2, 382) = 31.60, $p < .001$, $np^2 = .14$), blame attribution (F(2, 382) = 5.97, $p < .005$ $np^2 = .03$), and need for continued care (F(2, 382) = 18.63, $p < .001$, $np^2 = .09$). Consumption of standard alcohol drinks were consistently stigmatised more than those consuming alcohol-free drinks ($p$'s < .001). Consumption of standard alcohol drinks were stigmatised more than low alcohol drinks for continued care ($p < .0001$). Consumption of low alcohol

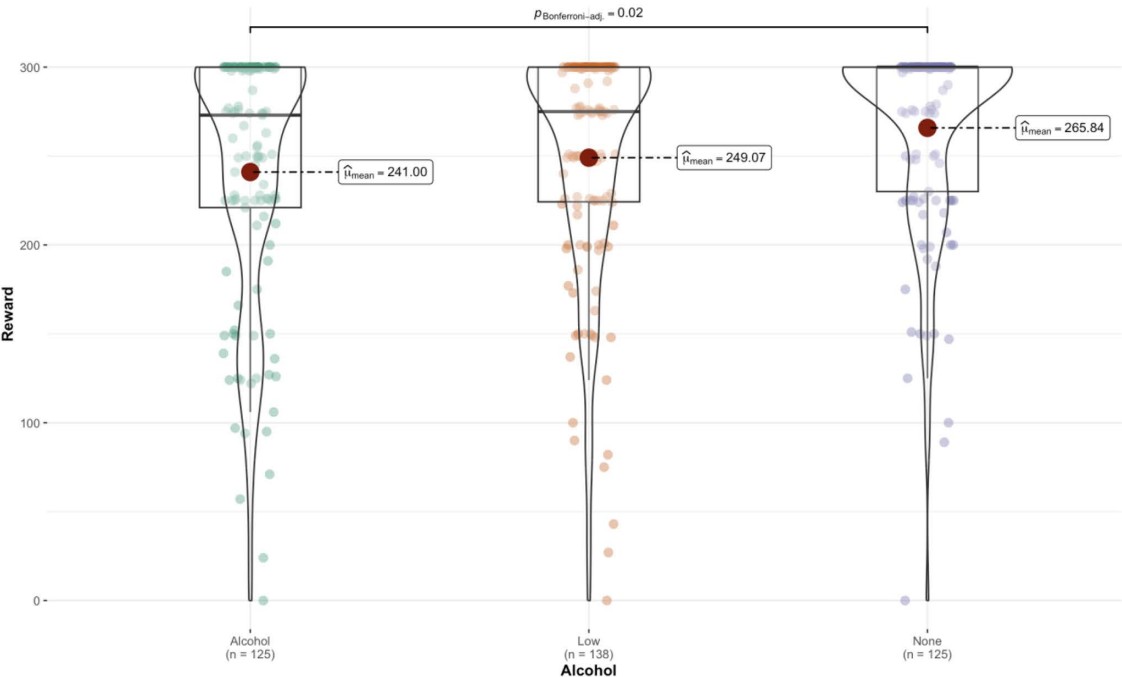

**Fig 1. Violin plot depicting the distribution of rewards assigned to individuals, displaying median values and individual data points across each alcohol condition.**

**Table 3. Tukey HSD results for the main effect of alcohol between subjects ANOVAs for study 1, 2 and 3. Presenting mean difference (95% confidence intervals),white=non-significant, red=<.05, yellow=<.01, orange=<.001,green=<.0001.**

| | Study 1 (n=179) | | | Study 2 (n=388) | | | Study 3 (n=487) | | |
|---|---|---|---|---|---|---|---|---|---|
| | Low-Standard | None-Standard | None-Low | Low-Standard | None-Standard | None-Low | Low-Standard | None-Standard | None-Low |
| Public stigma | 0.29 (−0.56, 1.14) | 0.03 (−0.85, 0.91) | −0.26 (−1.12, 0.60) | −1.06 (−1.77, −0.35) | −2.51 (−3.24, −1.78) | −1.45 (−2.16, −0.74) | −0.90 (−1.56, −0.24) | −1.76 (−2.42, −1.10) | −0.86 (−1.50, −0.22) |
| Treatment | 0.35 (−0.42, 1.11) | 0.21 (−0.59, 1.00) | −0.14 (−0.92, 0.64) | −0.13 (−0.65, 0.40) | −0.98 (−1.51, −0.44) | −0.85 (−1.37, −0.33) | −0.48 (−0.97, 0.01) | −0.89 (−1.38, −0.40) | −0.41 (−0.88, 0.06) |
| Personal dis-criminatory | −0.37 (−1.64, 0.90) | −0.34 (−1.66, 0.97) | 0.03 (−1.26, 1.31) | −1.05 (−2.26, 0.16) | −3.44 (−4.68, −2.20) | −2.39 (−3.60, −1.18) | −1.36 (−2.52, −0.20) | −2.32 (−3.47, −1.16) | −0.96 (−2.08, 0.16) |
| Personal stereotype | 0.45 (−1.37, 2.26) | 0.29 (−1.63, 2.20) | −0.16 (−2.01, 1.69) | −1.08 (−2.41, 0.24) | −3.36 (−4.71, −2.01) | −2.28 (−3.60, −0.95) | −1.64 (−2.79, −0.49) | −2.87 (−4.02, −1.73) | −1.23 (−2.34, −0.12) |
| Social distance | 2.00 (−2.74, 6.73) | 1.41 (−3.36, 6.19) | −0.59 (−5.30, 4.13) | 1.89 (0.09, 3.69) | 4.82 (2.97, 6.66) | 2.93 (1.13, 4.73) | 1.94 (0.40, 3.49) | 3.61 (2.07, 5.15) | 1.66 (0.17, 3.16) |
| Perceived danger | 0.04 (−3.82, 3.90) | −3.01 (−6.90, 0.88) | −3.05 (−6.89, 0.80) | −2.62 (−4.31, −0.93) | −5.94 (−7.67, −4.22) | −3.32 (−5.01, −1.63) | −2.01 (−3.62, −0.40) | −4.01 (−5.62, −2.41) | −2.01 (−3.56, −0.45) |
| Prognostic optimism | 2.38 (−1.78, 6.54) | 1.45 (−2.74, 5.65), | −0.93 (−5.07, 3.22) | 0.83 (−0.51, 2.17) | 4.36 (2.98, 5.74) | 3.53 (2.19, 4.87) | 0.94 (−0.33, 2.21) | 3.32 (2.05, 4.58) | 2.37 (1.15, 3.60) |
| Blame attribution | 0.40 (−1.46, 2.27) | 0.22 (−1.66, 2.10) | −0.18 (−2.04, 1.68) | −0.62 (−1.37, 0.12) | −1.12 (−1.88, −0.36) | −0.50 (−1.24, 0.25) | 0.12 (−0.64, 0.88) | −0.35 (−1.11, 0.40) | −0.47 (−1.20, 0.26) |
| Continued care | −0.23 (−0.89, 0.43) | −0.44 (−1.11, 0.22) | −0.21 (−0.86,.45) | −0.81 (−1.19, −0.44) | −0.90 (−1.29, −0.52) | −0.09 (−0.47, 0.28) | −0.19 (−0.58, 0.21) | −0.48 (−0.87, −0.09) | −0.30 (−0.68, 0.08) |

drinks were stigmatised more than alcohol free drinks on social distance ($p<.001$), perceived danger ($p<.0001$), and prognostic optimism ($p<.0001$)

### Study 3

**Social discrimination.** There was no significant main effect of alcohol type ($F(2, 481) = 1.50$, $p=.223$, $np^2=.01$) or social context ($F(1,481)=2.54$, $p=.112$, $np^2=.01$) on negative social response. There was no significant main effect of alcohol type ($F(2, 481) =.55$, $p=.580$, $np^2=.00$), see Fig 2, or social context ($F(1,481)=2.59$, $p=.108$, $np^2=.01$) on positive social response administered on the financial discrimination task.

**Denial of service.** Chi squared test demonstrated no significant associations between the alcohol type ($X2(2) = 0.85$, $p=.655$) or social context ($X2(2) = 0.43$, $p=.510$) and denial of service on the discrimination task.

**PPSM.** There was no significant effect of social context ($F(1, 481) =.02$, $p=.882$, $np^2=.00$) on PPSM total score, yet there was significant main effect of alcohol type ($F(2, 481) = 21.49$, $p<.001$, $np^2=.08$). Alcohol had a main effect on the PPSM subscales of public stigma ($F(2, 481) = 19.72$, $p<.001$, $np^2=.08$), treatment stigma ($F(2, 481) = 9.19$, $p<.001$, $np^2=.04$), personal discriminatory stigma ($F(2, 481) = 11.16$, $p<.001$, $np^2=.04$), and personal stereotype treatment stigma ($F(2, 481) = 17.37$, $p<.0001$, $np^2=.14$). Those who consumed standard alcohol drinks were consistently stigmatised more so than those consuming alcohol free drinks ($p's<.001$). Consuming standard alcohol drinks were stigmatised more so than consuming low alcohol drinks for public stigma ($p<.001$), personal discriminatory ($p<.05$) and personal stereotype ($p<.001$). Consuming low alcohol drinks were stigmatised more than alcohol free drinks on public stigma ($p<.001$) and personal stereotype ($p<.05$). For public stigma there was a significant interaction between alcohol type and social context

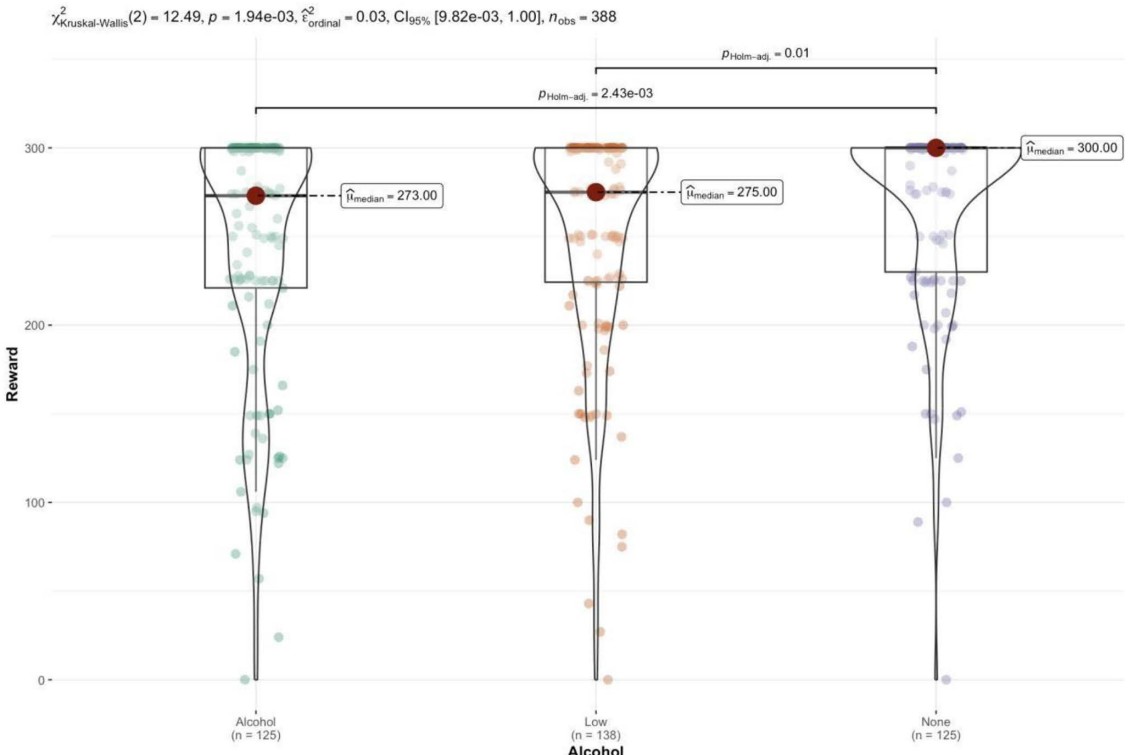

**Fig 2. Violin plot depicting the distribution of rewards assigned to individuals, displaying median values and individual data points across each alcohol condition.**

(F(2, 481) = 5.70, $p < .005$, $np^2 = .02$). In the alcohol condition perceived public stigma was higher in alone compared to social $(p < .001, d = .52)$, however there was no difference in the low $(p = .34, d = -.15)$ or the alcohol free conditions $(p = .23, d = .19)$ (see Fig 3).

**SAA.** There was no significant effect of social context (F(1, 481) = .62, $p = .433$, $np^2 = .00$) or alcohol type (F(2, 481) = 2.77, $p = .064$, $np^2 = .01$). on SAA total score. Alcohol type had a significant main effect on the SAA subscales of social distance (F(2, 481) = 15.15, $p < .001$, $np^2 = .06$), perceived danger (F(2, 481) = 17.32, $p < .001$, $np^2 = .07$), prognostic optimism (F(2, 481) = 20.66, $p < .001$, $np^2 = .08$), and need for continued care (F(2, 481) = 4.32, $p < .05$, $np^2 = .02$). Consumption of standard alcohol drinks was consistently stigmatised more so than consumption of alcohol free drinks $(p's < .05)$. Consumption of standard alcohol drinks were stigmatised more so than low alcohol drinks for social distance $(p < .01)$ and perceived danger $(p < .01)$. Consumption of low alcohol drinks were stigmatised more than alcohol free drinks on social distance $(p < .05)$, perceived danger $(p < .01)$ and prognostic optimism $(p < .0001)$. For social distance (F(2, 481) = 3.28, $p < .05$, $np^2 = .01$) and prognostic optimism distance (F(2, 481) = 4.35, $p < .05$, $np^2 = .02$) there is an interaction between alcohol type and social context.

### Pooled analysis across three studies

We pooled the effects of Social vs Alone (see Fig 4), as well as Alcohol vs Low Alcohol, Alcohol vs Alcohol free, Low alcohol vs Alcohol free (see Fig 5) across the three studies on the subscales of the SAA and PPSM. For social vs alone there were no significant pooled effects across the three studies. For Alcohol vs Alcohol free there were significant differences for social distance, prognostic optimism, perceived danger, public stigma and personal discriminatory scales. For alcohol vs low alcohol there were significant differences between social distance, prognostic optimism, danger and discrimination, whereby higher alcohol ABV% consumption was more stigmatised. Finally, for low alcohol vs alcohol free there were significant differences between blame, continued care, danger, public, stereotype and treatment, whereby low alcohol was more stigmatised.

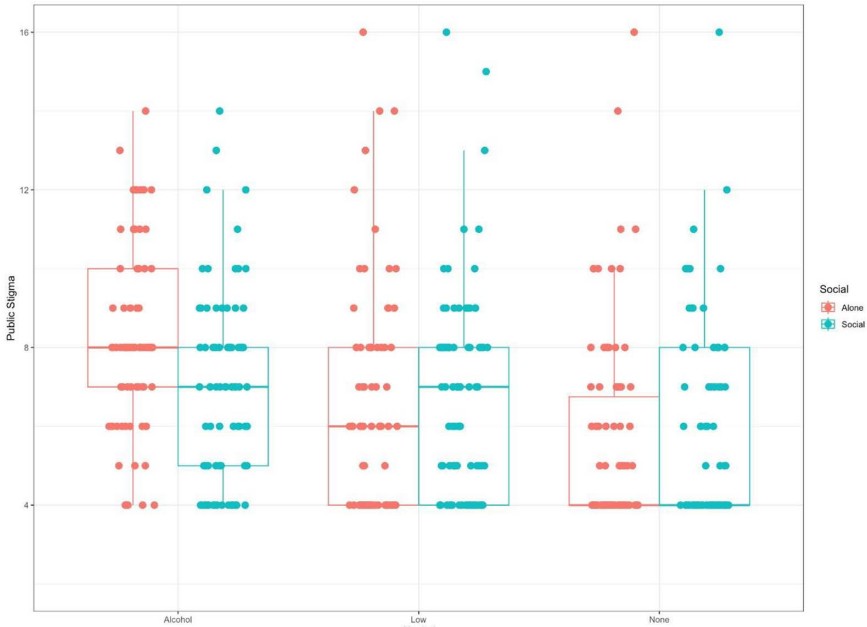

**Fig 3. Illustrates the levels of public stigma attributed to women consuming alcoholic, low-alcohol, and nonalcoholic beverages, stratified by social context (alone vs. social).** The box plot shows the median, interquartile range, and outliers.

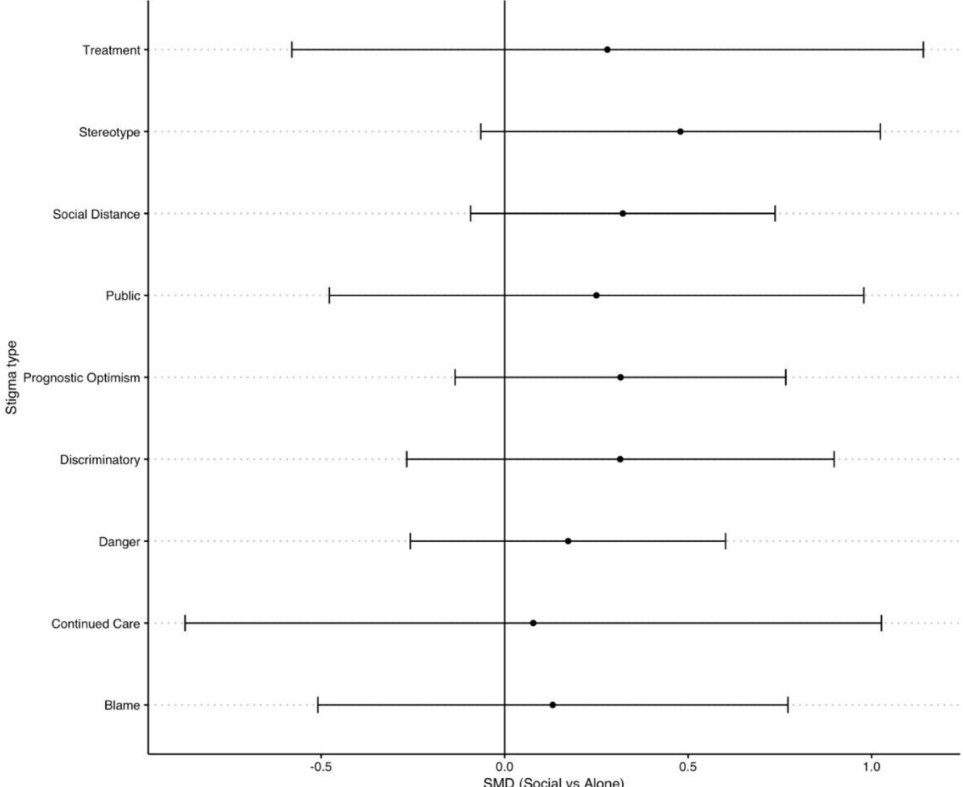

**Fig 4. Contrasts of Social vs Alone conditions on measures of stigma, pooled across the three studies.**

## Discussion

Across three studies, we aimed to explore whether individuals are more likely to stigmatise pregnant women who consume alcohol alone compared with socially and/or when consuming standard strength alcohol compared with lower strength and alcohol-free wine. Evidence supported the our hypothesis (H1.) as we demonstrated that stigma attribution was greater for hypothetical consumption of standard and lower strength alcohol products compared with alcohol free drinks, however evidence for a difference between low and standard strength alcohol-related stigma was inconsistent (only on public stigma, perceived danger, and social distance). There was limited evidence that social context of alcohol consumption impacted on stigma attribution or discriminatory behaviour, failing to support H2.

Our findings support previous research suggesting individuals hold stigmatising attitudes towards women who drink during pregnancy [7], perhaps due to the perceived risk to the fetus [8,49]. Our data suggests low alcohol drinks were still subject to some level of stigma in comparison to alcohol free products, particularly in respect to public stigma, personal stereotype, social distance, perceived danger, and prognostic optimism. This is potentially due to attitudes around drinking in pregnancy encompassing wider societal norms around expected abstinence during pregnancy, rather than the objective risk based on the strength of the alcohol [15] despite higher risk being associated with increasing alcohol strength [50], which is potentially based on inconsistent medical advice and poor knowledge in respect to NoLo products [25].

Our findings support the idea that stigma in this context, may be explained by the violation of the "motherhood ideal" [24]. This ideal promotes the expectation that mothers (including pregnant people) should prioritize their child's wellbeing above all else, and alcohol consumption, regardless of amount, is seen as antithetical to this expectation. Structural stigma, including institutional and societal attitudes particularly regarding continued care and reduced prognostic optimism

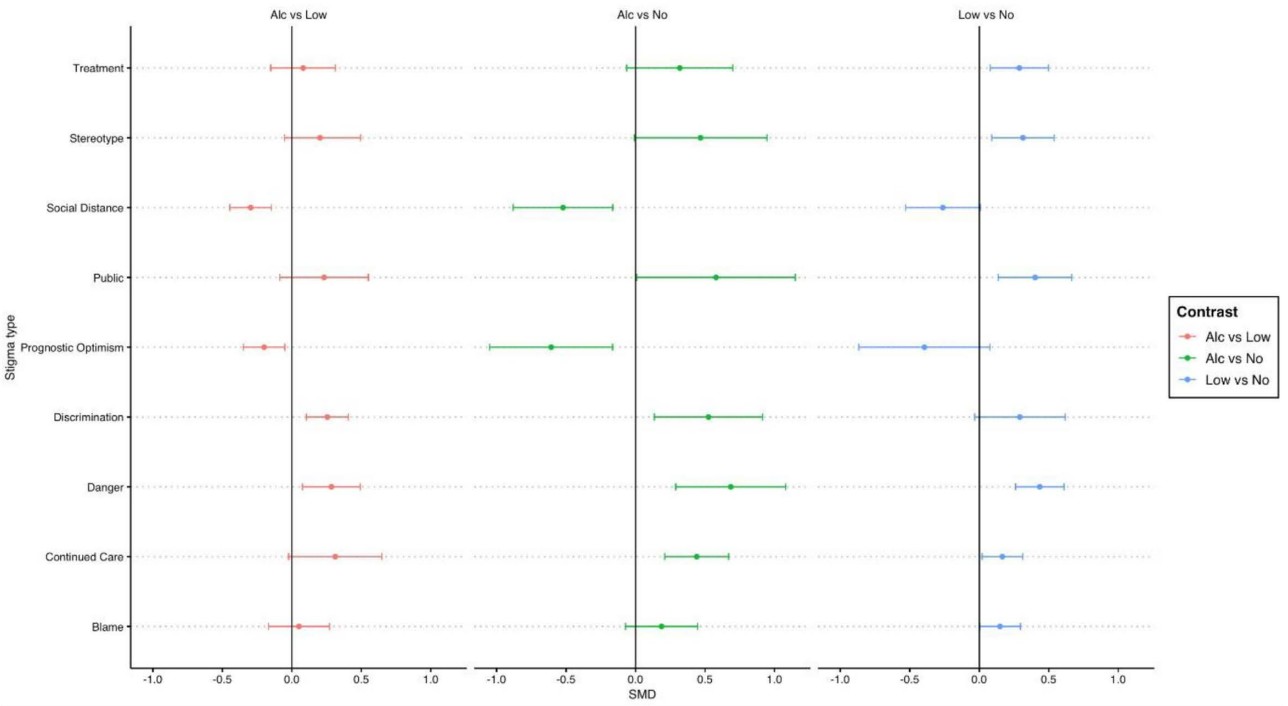

**Fig 5. Contrasts of Alcohol vs Low alcohol vs Alcohol free conditions on measures of stigma, pooled across the three studies.**

(expectation of a favourable outcome), exacerbates said issue by reinforcing punitive or judgmental views of people who drink while pregnant [13]. Overly stigmatising attitudes may discourage pregnant women from seeking help for alcohol-related issues [26]. In contrast a more positive framing around abstinence during pregnancy, which acknowledges individual difficulties, may improve public health messaging and improve female health outcomes [51].

The attribution of blame also plays a critical role in the stigmatisation process. Participants were more likely to attribute personal blame to women consuming standard alcohol than to those consuming lower-alcohol- or alcohol-free beverages. This aligns with Matthews (2019), which suggests that blame is more likely to be assigned in cases where individuals are perceived to have control over their behaviour. This is particularly salient in respect to alcohol use in pregnancy where internal self-stigma (e.g., shame and guilt) may drive alcohol use [28,52]. However, this fails to account for the complex social and psychological factors that influence alcohol consumption during pregnancy, such as mental health issues, health literacy, stability of substance use, and social pressures [25,27,53].

There are some limitations to the series of studies presented. First, the vignette method employed was hypothetical and may lack ecological validity, but provides proof-of-concept for future investigations, such as ecological momentary assessment paradigms to capture lived experience (Jones[54] 2024). Second, our samples within the study are heavily skewed, future work should seek to examine the effects of stigma accounting for gender disparity. Views of partners and other support figures should also be garnered given the role of men's on influencing women's alcohol use during pregnancy [55]. In addition, differences in age and missing educational data, particularly in Study 1, limit the comparability of pooled effects across studies and may affect the generalisability of findings, warranting cautious interpretation and more demographically balanced sampling in future research.. Future research should see to investigate this further particularly given the variability in stigma perceptions across age and education levels [56]. The measure of stigma used may not be the most appropriate measure of stigma for this population of interest. Future research should endeavour to

use more nuanced scale that capture this unique period and specifically related to alcohol, given prior work has cited the lack of scales for this specific population and substance use [57]. The study relied solely on quantitative methods, which, while valuable for identifying patterns of stigma attribution, do not provide insights into the subjective reasoning behind participants' judgments. Qualitative investigation would allow for a more detailed and nuanced account of stigmatisation capturing moral, cultural and experiential factors that shape said stigma. Finally, our study did not consider the intersectional nature of stigma. Disproportionately high levels of stigma are experienced by those from lower socio-economic background and minority ethnic groups [13], including heightened alcohol-based stigma towards those from marginalised groups [58]. Future research should investigate the intersectionality of stigma which will allow for more tailored publish health messaging, that accounts for women's individual circumstances, e.g., ethnicity and socio-economic status [59]. To fully explore such aspects of stigma and

## Conclusions

To summarise, across three studies we demonstrated that women were stigmatized for consuming alcohol during pregnancy, with higher stigma levels attributed to those consuming standard or low-strength alcohol compared to non-alcoholic drinks. Although there was some evidence that strength of alcohol may influence certain aspects of stigma, this needs to be confirmed by future research utilising more ecologically valid methods. The social context of alcohol consumption had limited impact on stigma, suggesting that societal expectations around abstinence in pregnancy play a stronger role than the social context in which alcohol is consumed. Policy should seek to promote and expedite support for alcohol use during pregnancy, given prior work has shown the positive effect of pregnancy disclosure or substance treatment has on women [26]

## Supporting information

**S1 File. Supplementary materials.**
(DOCX)

## Informed consent

Informed consent was obtained from all individual adult participants included in the study.

## Author contributions

**Conceptualization:** Sam Burton, Andrew Jones.

**Data curation:** Sam Burton, Shannon Cummings, Andrew Jones.

**Formal analysis:** Sam Burton, Andrew Jones.

**Investigation:** Sam Burton, Shannon Cummings, Charlotte Connides-Smith, Elizabeth Fairweather, Poppy Whalley, Andrew Jones.

**Methodology:** Sam Burton, Andrew Jones.

**Project administration:** Sam Burton, Andrew Jones.

**Resources:** Sam Burton, Andrew Jones.

**Software:** Andrew Jones.

**Supervision:** Sam Burton.

**Validation:** Sam Burton, Andrew Jones.

**Visualization:** Sam Burton, Andrew Jones.

**Writing – original draft:** Sam Burton, Shannon Cummings, Charlotte Connides-Smith, Elizabeth Fairweather, Catharine Montgomery, Abigail K Rose, Poppy Whalley, Andrew Jones.

**Writing – review & editing:** Sam Burton, Shannon Cummings, Charlotte Connides-Smith, Elizabeth Fairweather, Catharine Montgomery, Abigail K Rose, Poppy Whalley, Andrew Jones.

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
