## [Decision Letter · Decision Letter 0]

2 Oct 2025

Dear Dr. Sam Burton,

Thank you for submitting your manuscript to PLOS ONE. After careful consideration, we feel that it has merit but does not fully meet PLOS ONE’s publication criteria as it currently stands. Therefore, we invite you to submit a revised version of the manuscript that addresses the points raised during the review process.

We look forward to receiving your revised manuscript.

Kind regards,

Martin Mbonye, PhD

Academic Editor

PLOS ONE

Reviewers' comments:

Reviewer's Responses to Questions

**Comments to the Author**

1. Is the manuscript technically sound, and do the data support the conclusions?

Reviewer #1: Partly

Reviewer #2: Partly

2. Has the statistical analysis been performed appropriately and rigorously?

Reviewer #1: Yes

Reviewer #2: I Don't Know

3. Have the authors made all data underlying the findings in their manuscript fully available?

Reviewer #1: Yes

Reviewer #2: Yes

4. Is the manuscript presented in an intelligible fashion and written in standard English?

Reviewer #1: Yes

Reviewer #2: Yes

Reviewer #1: This study addresses an important topic—the stigma associated with alcohol consumption during pregnancy—through three questionnaire-based sub-studies. While the research question is of clear societal and public health relevance, both the conceptual framing and the study design appear insufficiently developed.

Methodological concerns:

• It is not clear why stigma perceptions were assessed in relation to women consuming non-alcoholic products. The finding that such behavior is associated with lower stigma is highly predictable and appears to provide little added value.

• The stigma questionnaire does not seem well-suited for use with pregnant women, particularly the item(s) related to perceived dangerousness.

• In Study 1, participants were considerably younger than in the subsequent sub-studies, and the majority did not provide information on educational background. The implications of these heterogeneous samples for the comparability and interpretation of results require closer examination.

• Among the 1,054 participants, only 33 were men. It remains unclear why the study was unable to recruit a more balanced gender distribution. The potential influence of this imbalance on the findings should be explicitly discussed.

Additional issues:

• The referencing is at times inappropriate. For readers familiar with the literature, it becomes evident that some cited sources are not referenced in relation to their primary outcomes, while at other points they are mentioned in ways that do not accurately reflect the actual findings. This requires careful revision to ensure accurate and transparent use of sources.

Reviewer #2: Thank you for taking time to work on this manuscript. It is an important topic that can be contribute to the well being of maternal health. Below are a few comments that I feel can make your manuscript strong. These are provided in an attachment separately.

**Do you want your identity to be public for this peer review?** For information about this choice, including consent withdrawal, please see our Privacy Policy

Reviewer #1: **Yes: ** Annette Binder

Reviewer #2: No

---

## [Author Response · Author response to Decision Letter 1]

18 Nov 2025

Reviewer #1: This study addresses an important topic—the stigma associated with alcohol consumption during pregnancy—through three questionnaire-based sub-studies. While the research question is of clear societal and public health relevance, both the conceptual framing and the study design appear insufficiently developed.

Methodological concerns:

• It is not clear why stigma perceptions were assessed in relation to women consuming non-alcoholic products. The finding that such behavior is associated with lower stigma is highly predictable and appears to provide little added value.

Response: We thank the reviewer for this comment and agree this could have been made clearer, we have added a brief insertion to recent qualitative work that discusses how women still experience shame and guilt when consuming NoLo products in certain situations. Our qualitative findings (Burton et al., 2025) demonstrate that NoLo drinks serve not only as alcohol substitutes but as tools for social inclusion, concealment of early pregnancy, and stigma management, underscoring the need to understand the degree, not just the presence, of stigma across different beverage categories. By including non-alcoholic products in our stigma comparisons, we provide critical insight into how social judgment operates along a continuum, rather than as a binary (drinking vs. not drinking), helping to inform public health messaging, policy, and harm-reduction strategies.

‘Given the increasing marketisation of NoLo products (Nicholls, 2023) it is crucial to understand the stigma attributed towards their use, given how recent work suggests women may still experience guilt and shame following consumption of NoLo products, and that such products aid in management of stigmatisation but in turn may also create stigma by appearing as standard alcohol products (Burton et al., 2025).’

• The stigma questionnaire does not seem well-suited for use with pregnant women, particularly the item(s) related to perceived dangerousness.

Response: We agree with the reviewer and have highlighted this within the limitations section, along with prior work that has also stated this is an issue. We are currently developing a specific scale for alcohol related stigma during the perinatal period. We have also highlighted within the limitations how qualitative work may capture such stigmatising behaviour in more detail than quantitative measures.

‘The measure of stigma used may not be the most appropriate measure of stigma for this population of interest. Future research should endeavour to use more nuanced scale that capture this unique period and specifically related to alcohol, given prior work has cited the lack of scales for this specific population and substance use (Bann et al., 2023).’

‘The study relied solely on quantitative methods, which, while valuable for identifying patterns of stigma attribution, do not provide insights into the subjective reasoning behind participants’ judgments. Qualitative investigation would allow for a more detailed and nuanced account of stigmatisation capturing moral, cultural and experiential factors that shape said stigma’

• In Study 1, participants were considerably younger than in the subsequent sub-studies, and the majority did not provide information on educational background. The implications of these heterogeneous samples for the comparability and interpretation of results require closer examination.

Response: We thank the reviewer for highlighting this important point regarding the heterogeneity in participant demographics across the three studies. We agree that the younger age profile and incomplete educational data in Study 1, compared to Studies 2 and 3, introduce potential concerns regarding the comparability and generalisability of findings across samples. As such, part of our rationale in terms of analysis was to analyse studies separately and combined, to account for such differences. We do highlight this as a limitation within our discussion for the reader

‘In addition, differences in age and missing educational data, particularly in Study 1, limit the comparability of pooled effects across studies and may affect the generalisability of findings, warranting cautious interpretation and more demographically balanced sampling in future research. Future research should see to investigate this further particularly given the variability in stigma perceptions across age and education levels (Sudhinaraset, Wigglesworth, & Takeuchi, 2016) ‘

We also state within the participants section that study 1 was used to power subsequent studies:

‘. In study 1 we recruited 179 participants; sample size was not pre-defined a priori but we used the effects from study 1 to power subsequent studies’

• Among the 1,054 participants, only 33 were men. It remains unclear why the study was unable to recruit a more balanced gender distribution. The potential influence of this imbalance on the findings should be explicitly discussed.

Response: We used a crowdsourcing platform to recruit participants, and did not actively exclude men from the sample, as such fewer men were recruited by chance. We do discuss within our limitations that the sample is heavily skewed, and future work should account for gender disparities. We have also expanded our discussion to emphasize the importance of understanding perspectives across genders and within diverse social roles (including partners and informal support networks) given their collective influence on alcohol-related norms and behaviours during pregnancy

‘Second, our samples within the study are heavily skewed, future work should seek to examine the effects of stigma accounting for gender disparity. Finally, our study did not consider the intersectional nature of stigma. Views of partners and other support figures should also be garnered given the role of men’s on influencing women’s alcohol use during pregnancy (Dimova et al., 2022). ’

Additional issues:

• The referencing is at times inappropriate. For readers familiar with the literature, it becomes evident that some cited sources are not referenced in relation to their primary outcomes, while at other points they are mentioned in ways that do not accurately reflect the actual findings. This requires careful revision to ensure accurate and transparent use of sources.

Response: We have reviewed the manuscript and made amendments accordingly, we thank the reviewer for this comment.

Reviewer 2:

Thank you for taking time to work on this manuscript. It is an important topic that can be contribute to the wellbeing of maternal health. Below are a few comments that I feel can make your manuscript strong.

Abstract

1. The way you have started the introduction of your abstract seems lacking. For example, you do not state what the problem is and why it is important but you dive straight to examining a problem that is not well articulated. Please introduce your study well

Response: We have added a sentence to try to briefly set the problem as to why this is a public health concern, while staying within the word limit for the abstract.

‘Alcohol use during pregnancy remains a major public health concern, with societal expectations of abstinence often resulting in stigma toward women who drink. To examine stigma and discrimination attributed to pregnant women consuming alcohol of varying strength and social context, within hypothetical scenarios. We explored whether greater stigma was attributed to consuming alcohol in social settings vs. alone, and whether consumption of low and alcohol-free drinks are less stigmatised than standard strength alcohol drinks.’

2. In the methodology section, you do not state precisely how your data was analysed and the software used. Please review and make it clear.

Response: We have stated that the data was analysed using R studio and specify packages used outside of base R, within the methodology section. We have sought to clarify for the reader what the dependent variables were to increase clarity.

‘Data were analysed using R studio with the ‘tidyverse’, ‘’ggplot’, ‘effectsize’, and ‘psych’ packages. Any income greater than what was deemed implausible by examination of the distributions and box plot was also removed (10 participants reporting income of above £270,000). These participants were retained for the inferential analysis. Between subject ANOVAs were used to test for main effects and interactions in terms of social context (2 levels: alone or with friends) and alcohol type (3 levels: no, low and standard alcohol), in all studies for the dependent variables of SAA, PPSM and financial discrimination task (for study 2 and 3).’

3. I assume these are your results: Pregnant women consuming standard and low-alcohol drinks were more stigmatised than those consuming alcohol-free beverages across multiple aspects of stigma. Stigma attribution also varied between low and standard alcohol consumption conditions yet were inconsistent. You have not guided the reader to follow how consuming alcohol was more stigmatizing and also how attribution varied. This would benefit the reader if you drew some statistical expressions of this rather than just writing a test that is not well substantiated. This needs to be rewritten and present the results well.

Response: We have amended the results aspect of the abstract as follows.

‘Pregnant women consuming standard and low-alcohol drinks were more stigmatised than those consuming alcohol-free beverages across multiple aspects of stigma (p < .05). Pooled analyses showed that standard alcohol was associated with greater stigma on measures including social distance, perceived danger, prognostic optimism, public stigma, and personal discrimination. Low-alcohol drinks were also more stigmatised than alcohol-free drinks across domains such as blame, continued care, public stigma, and treatment stigma.’

4. What is the implication of your study/ its contribution to policy and practice? I think the most important question to answer her is so what from what you have presented.

Response: We have added a brief insertion while remaining conscious of the word limit for this section.

‘Public health messaging should shift towards risk-informed education to shape societal norms of abstinence and reduce stigmatisation.’

Main body

Introduction:

5. In the second paragraph of the introduction, you state: In order to develop more effective, tailored alcohol interventions to reduce alcohol exposed pregnancy, it is also important to understand why women do not drink and why they do not seek help for alcohol use during pregnancy. This statement is confusing. Are you trying to explore why pregnant women do not drink alcohol or why they drink? Please review this and make it clear on what you are trying to bring to the attention of the reader.

Response: We have sought to clarify this for the reader and changed do not drink to abstinence. From a public health perspective it is important to know why women do and don’t drink, but also drivers of both behaviour choices to inform person centred policy.

‘In order to develop more effective, tailored alcohol interventions to reduce alcohol exposed pregnancy, it is also important to understand why women remain abstinent during pregnancy and why they do not seek help for alcohol use during pregnancy (Fleming, Gomez, Goodwin, & Rose, 2023).’

6. You state standard alcohol drinks but you have not defined what this means and even give some examples for people to understand.

Response: We now define this in terms of percentage of alcohol in the first instance and have worded this as typical to avoid confusion across different countries where the measurement of a unit may differ.

‘While drinking typical alcohol drinks (above 1.2% ABV, e.g. not a low or no alcohol drink) may be stigmatised and unacceptable during pregnancy’

7. In the introduction, there is where you mention “NoLo drinks”. Please define this in the context of the presentation so that people can associate with it.

Response: We draw the reviewer’s attention to this section of the introduction, where we have defined what these products are.

‘While drinking standard alcohol drinks may be stigmatised and unacceptable during pregnancy (Binder et al., 2024; Culp et al., 2022; Weber, Miskle, Lynch, Arndt, & Acion, 2021), recent advancements in the alcohol-free beverage market (Nicholls, 2023) require further investigation in respect to stigma and acceptability during pregnancy. NoLo products may be deemed as more acceptable, we must understand the wider impacts given there is no known safe level of alcohol consumption (Dejong, Olyaei, & Lo, 2019). Currently there are several categories of ‘NoLo’ drink products, e.g., alcohol-free (≤0.05% ABV), de-alcoholised (0.05-0.5% ABV), and low alcohol (≤1.2% ABV). The alcohol industry is also diversifying with an increase in alcohol products with a lower alcohol beverage volume (ABV) than the ‘standard’(e.g. 1.2% ABV or less) (Care, 2018). Lower than standard ABV drinks may be consumed as a proxy for NoLo drinks rather than abstinence(Pardoe, 2020), this is primarily due to poor knowledge around categorisation definitions(Okaru & Lachenmeier, 2022). Within the context of pregnancy, NoLo and lower than average alcohol products may be stigmatised less than standard products, which in turn may enable alcohol use against abstinence recommendations (Goh, Verjee, & Koren, 2010). ‘

Methods:

8. Where was this study conducted? Please state the date of data collection.

Response: We now specify that participants are recruited from the UK and it already states dates of data collection at the end of the paragraph.

‘Across the three studies, 1054 participants were recruited from the UK to take part in an online study, see Table 1 for demographics. In study 1 we recruited 179 participants; sample size was not pre-defined a priori but we used the effects from study 1 to power subsequent studies. Study 2 was powered based off simulation analysis for a 2x3 between subjects design, using Superpower package in R. Based off 90% power, alpha=0.05 it was estimated we would need 64 participants per group (target N=384, actual N=388). In study 3 we powered based on simulation analysis of a main effect of alcohol on the financial discrimination task in study 2. We estimated with 90% power, alpha=0.05 we would need 81 participants per group (target N=486, actual N=487). Recruitment began on the 24/2/2024 and ended on the 4/11/2024’

9. How was data collected and who collected it? Was it a self-administered questionnaire or there was somebody who was taking the participant through? This is not clearly stated and needs to be made clear.

Response: We now specific within the design and procedure this was self administered.

‘In each study participants were instructed to complete the experiment in a quiet space without distractions (self administered).’

10. Were your participants only women? If so, do you think that you missed important views from the men that would have informed your study? I realise in your demographics, you had around 11. Please give reasons why this sample is so small and basically the rationale for these 11.

Response: We used a crowdsourcing platform to recruit participants, and did not actively exclude men from the sample, as such minimal men were recruited by chance. We do discuss within our limitations that the sample is heavily skewed and future work should account for gender disparities.

‘Second, our samples within the study are heavily skewed, future work should seek to examine the effects of stigma accounting for gender disparity. Finally, our study did not consider the intersectional nature of stigma.’

11. Best practice is to define an abbreviation the first time you use it. For example answer the SSA and PPSM these terms are first presented as abbreviations and later put in full down. Please rectify this.

Response: We have now rectified this.

12. Stigma and Attribution Assessment (SSA): is this a conventionally accepted abbreviation? If not, why not SAA? You will need to explain this so that we are on the same page.

Response: This has now been changed to SAA throughout, this was a typographical error.

Results section

13. In study 2, you state: Consumption of standard alcohol drinks were consi

---

## [Decision Letter · Decision Letter 1]

16 Dec 2025

The stigma of alcohol use during pregnancy: Exploring the impact of alcohol strength and social context on public perceptions

PONE-D-25-19244R1

Dear Dr. Burton,

We’re pleased to inform you that your manuscript has been judged scientifically suitable for publication and will be formally accepted for publication once it meets all outstanding technical requirements.

Kind regards,

Martin Mbonye, PhD

Academic Editor

PLOS One

Additional Editor Comments (optional):

Reviewers' comments:

Reviewer's Responses to Questions

**Comments to the Author**

Reviewer #1: All comments have been addressed

Reviewer #2: All comments have been addressed

2. Is the manuscript technically sound, and do the data support the conclusions?

Reviewer #1: Yes

Reviewer #2: Yes

3. Has the statistical analysis been performed appropriately and rigorously?

Reviewer #1: Yes

Reviewer #2: Yes

4. Have the authors made all data underlying the findings in their manuscript fully available?

Reviewer #1: Yes

Reviewer #2: Yes

5. Is the manuscript presented in an intelligible fashion and written in standard English?

Reviewer #1: Yes

Reviewer #2: Yes

Reviewer #1: Thank you for the opportunity to review the manuscript again. The revisions have significantly improved its clarity. The points I had previously raised were addressed appropriately by the authors or have now been explained in a convincing manner. I therefore support its publication.

Reviewer #2: (No Response)

**Do you want your identity to be public for this peer review?** For information about this choice, including consent withdrawal, please see our Privacy Policy

Reviewer #1: **Yes: ** Annette Binder

Reviewer #2: No

---

## [Editor Report · Acceptance letter]

PONE-D-25-19244R1

PLOS One

Dear Dr. Burton,

I'm pleased to inform you that your manuscript has been deemed suitable for publication in PLOS One. Congratulations! Your manuscript is now being handed over to our production team.

Kind regards,

on behalf of

Dr. Martin Mbonye

Academic Editor

PLOS One